# Multiple Roles of SMC5/6 Complex during Plant Sexual Reproduction

**DOI:** 10.3390/ijms23094503

**Published:** 2022-04-19

**Authors:** Fen Yang, Ales Pecinka

**Affiliations:** 1Centre of the Region Haná for Biotechnological and Agricultural Research (CRH), Institute of Experimental Botany (IEB), Czech Academy of Sciences, 77900 Olomouc, Czech Republic; yang@ueb.cas.cz; 2Department of Cell Biology and Genetics, Faculty of Science, Palacký University, 77900 Olomouc, Czech Republic

**Keywords:** SMC5/6 complex, genome stability, meiosis, seed, reproductive development, fertility, polyploidy

## Abstract

Chromatin-based processes are essential for cellular functions. Structural maintenance of chromosomes (SMCs) are evolutionarily conserved molecular machines that organize chromosomes throughout the cell cycle, mediate chromosome compaction, promote DNA repair, or control sister chromatid attachment. The SMC5/6 complex is known for its pivotal role during the maintenance of genome stability. However, a dozen recent plant studies expanded the repertoire of SMC5/6 complex functions to the entire plant sexual reproductive phase. The SMC5/6 complex is essential in meiosis, where its activity must be precisely regulated to allow for normal meiocyte development. Initially, it is attenuated by the recombinase RAD51 to allow for efficient strand invasion by the meiosis-specific recombinase DMC1. At later stages, it is essential for the normal ratio of interfering and non-interfering crossovers, detoxifying aberrant joint molecules, preventing chromosome fragmentation, and ensuring normal chromosome/sister chromatid segregation. The latter meiotic defects lead to the production of diploid male gametes in Arabidopsis SMC5/6 complex mutants, increased seed abortion, and production of triploid offspring. The SMC5/6 complex is directly involved in controlling normal embryo and endosperm cell divisions, and pioneer studies show that the SMC5/6 complex is also important for seed development and normal plant growth in cereals.

## 1. Introduction

Recent studies revealed extensive dynamics in plant nuclear and chromosomal organization [1,2,3]. Plant chromosome and chromatin are very dynamic during sexual reproduction, where different stages are quickly substituted for one another and many events occur in a small number of highly specialized cells [4,5,6]. The process starts with meiosis, continues through gametogenesis and fertilization, and finally seed development. Sexual reproduction is crowned by precisely controlled nuclear divisions, the regulated repair of programmed DNA double-strand breaks, and a reduction in chromosome number. Many of these steps are orchestrated and surveilled by the family of evolutionarily conserved large-scale chromatin-processing molecular machines called structural maintenance of chromosomes (SMCs), along with other factors [7,8]. Whereas the functional role of cohesin and condensin SMC complexes during plant reproductive development has long been known [9], the functions of the SMC5/6 complex emerged only recently and are quickly accumulating [10]. The SMC5/6 complex is known mainly for its role during DNA damage repair across eukaryotic kingdoms [11,12,13]. In plants, it has been identified as a factor required for normal levels of somatic homologous recombination (HR) and resistance to various types of DNA-damaging agents [10,14]. However, recent papers on plant reproduction revealed many new fascinating phenotypes of SMC5/6 complex mutants. Furthermore, these studies also contributed strong evidence suggesting that this complex affects multiple processes within one developmental stage, sometimes via unknown or previously unanticipated pathways. Therefore, our review focuses mostly on the new findings concerning SMC5/6 functions in plant sexual reproductive development.

The basic subunits and structure of the SMC5/6 complex have been described [13,15,16]. Briefly, it consists of two SMC proteins and up to six (organism-dependent) additional subunits, commonly termed Non-SMC elements (NSEs). The SMC5 and SMC6 proteins are ATPases that form a heterodimer via their hinge domains. At the opposite end, the SMC heads are bridged via the kleisin-type protein NSE4. The NSE1 and NSE3 subunits bind the NSE4 and are important for the complex’s attachment to DNA and processivity. The targets of ubiquitin ligase NSE1 [17] remain unknown, but a recent study revealed that it can ubiquitinate NSE4 in *Schizosaccharomyces pombe* [18]. The NSE2 (a.k.a. Methyl methanesulfonate-sensitive 21-MMS21, HIGH PLOIDY 2-HPY2) is an E3 small ubiquitin modifier (SUMO) ligase attached to the coiled-coil of SMC5 [19,20,21,22]. Its SUMOylation targets likely differ between organisms and include DNA repair proteins, as well as some SMC5/6 complex subunits [19,23,24]. The only identified NSE2 target in Arabidopsis is NSE4A [25]. In addition, there are two to three evolutionarily non-conserved subunits, generally thought to facilitate the complex’s binding to a specific chromatin context. In yeasts, Nse5 and Nse6 t interact with BRCT domain-containing factors Rtt107 or Brc1 [26,27,28]. In humans, there are SMC5-SMC6 complex localization factors 1 and 2 (SLF1 and SLF2) [29]. In plants, the functional homologs of NSE5 and NSE6 are ARABIDOPSIS SNI1-ASSOCIATED PROTEIN 1 (ASAP1) and SUPPRESSOR OF NPR1; INDUCIBLE (SNI1), respectively [30,31].

## 2. Essential Functions in Meiosis

Meiosis is a specialized division that occurs in sexually reproducing eukaryotes and fulfills two important tasks [32]. First, it facilitates a controlled exchange of segments of homologous chromosomes, leading to new combinations of genetic information. Second, with the two subsequent rounds of cell division, it produces four haploid gametes from one meiocyte. The most complex part of meiosis is prophase I, which starts with the leptotene stage, when numerous DNA double-strand breaks (DSBs) are introduced along the chromosomes by SPORULATION 11 (SPO11). Single-stranded DNA overhangs produced by MRN (MRE11-RAD50-NBS1) complex [33] are bound by two DNA recombinases, RAD51 and DMC1, invade appropriate regions on the homologous chromosomes, and form D-loop intermediates. Several repair pathways processing D-loop intermediates have been described in plants (Figure 1).

The intermediates forming double Holliday junctions (dHJs) can either be removed via RTR (RECQ4A-TOP3A-RMI1) pathway, forming non-crossovers (NCOs) by dissolution [32,34,35], or processed into the Class I crossovers (COs) via ZMM (ZYP-MER3-MSH) pathway by dHJ resolution [32,36]. In contrast, the intermediates forming single Holliday junctions (sHJ) are processed either as the Class II crossovers by the MMS AND UV SENSITIVE 81 (MUS81) pathway or as NCOs by synthesis-dependent strand annealing (SDSA) HR pathway [32,34]. Subsequently, the physical connection formed by COs (chiasmata) keeps bivalents together to ensure proper orientation and segregation of chromosomes during the first meiotic division. During meiosis II, sister chromatids segregate, and four haploid spores are formed. All four male products survive, whereas three of the female products are eliminated.

**Figure 1 ijms-23-04503-f001:**
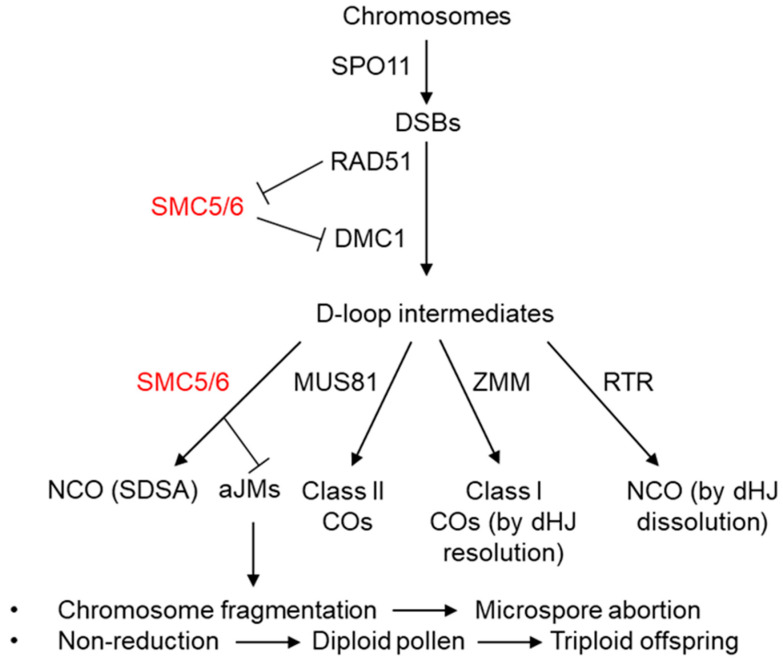
Simplified model of the functions of SMC5/6 complex during male meiosis in Arabidopsis. The meiotic double-strand breaks (DSBs) are initiated by SPO11. Recombinases RAD51 and DMC1, are required to repair the breaks by homology search and strand invasion, respectively. RAD51 supports the action of DMC1 by attenuating the SMC5/6 complex during this step [37]. The invasion of the homologous duplex DNA gives rise to a D-loop intermediate, which can be repaired by (i) non-crossovers (NCOs) via double Holliday junction (dHJ) dissolution by the RTR complex; (ii) by the ZMM pathway, forming Class I crossovers (COs) via dHJ resolution; (iii) by the MUS81-dependent pathway, forming Class II Cos; or (iv) by synthesis-dependent strand annealing (SDSA), generating NCOs. SMC5/6 is required to prevent the accumulation of aberrant and unresolved intermediates that arise outside the ZMM pathway [38]. The absence of SMC5/6 complex functions leads to the accumulation of abnormal joint molecules (aJMs), which are either repaired by MUS81, promoting Class II COs [39], or result in different abnormal phenotypes: either chromosome fragmentation and microspore abortion or lack of chromosome segregation, which results in the formation of diploid gametes and, finally, triploid offspring [40].

The involvement of the SMC5/6 complex in meiosis has been reported across kingdoms. Studies based on yeast and animal models suggested its role in the removal of unregulated (abnormal) joint molecules (aJMs) that may appear among sHJs [38,41,42,43]. This is also supported by observations in plants. Analysis of Arabidopsis natural variation in CO frequencies revealed a higher degree of recombination in L*er* compared to Col-0 accession [39]. Subsequent quantitative trait locus (QTL) mapping led to the identification of the natural allele of *SNI1* as the causal gene. Natural variation in SNI1 might fit well with its putative function as a loader to chromatin [31], and such a natural allele could modulate binding to slightly different substrates and/or under different conditions. The database of >1000 Arabidopsis genomes displays considerable variation in SNI1, with more than 30 alleles and 59 amino acid substitutions throughout the entire protein. The most likely causal mutation in SNI1^L*er*^ is I235V. Currently, it is unknown how this substitution changes SNI1 function, but valine represents only a minor change in chemical properties compared to isoleucine. Phenotypes similar to SNI1^L*er*^ (although stronger) were observed in loss-of-function SMC5/6 complex mutants *sni1-2*, *nse4a-2,* and *asap1* (Table 1). When looking at the COs, it became obvious that Arabidopsis SMC5/6 complex mutants show less CO interference, and the relative amount of Class I COs decreased compared to Class II COs. Collectively, this suggests that the SMC5/6 complex is involved in aJM resolution during meiotic prophase I in Arabidopsis. An absence of this activity leads to the formation of anaphase bridges, chromosome fragmentation, unequal segregation, and chromatin loss. The SMC5/6 complex could eliminate aJMs by least two (mutually non-exclusive) mean. First, it might stabilize the HR structure and provide operational space for the other repair factors, including RMR complex, MUS81, and/or cohesin. This has been experimentally proven in yeast and mammals [38,41,44]. In Arabidopsis, there are, so far, only data from somatic nuclei, where the homologous chromosome regions are significantly less associated upon DNA damage in SMC5/6 complex mutants [45]. Secondly, the dynamics of aJM repair might be affected by the SUMOylation activity of the SMC5/6 complex. However, this has not been explored in plants.

Although the activity of the SMC5/6 complex is important for aJM resolution, it needs to be correctly timed during meiosis. A recent study in Arabidopsis revealed that the SMC5/6 complex has to be suppressed during the early stages of prophase I to allow for the efficient formation of dHJs [37]. This was found by a forward-directed screen for suppressors of *sni1*-induced sterility. The screen yielded *sni1 rad51* double mutant, suggesting that SMC5/6 complex and RAD51 might have antagonistic roles during meiosis. RAD51 is a homolog of bacterial RecA, which is protein known to bind single-stranded DNA (ssDNA) and is essential for sequence homology search. In meiosis, RAD51 cooperates with another recA homolog, DISRUPTION OF MEIOTIC CONTROL 1 (DMC1). Although RAD51 performs homology search, the DMC1 is specialized in strand invasion. Loss of function from each of these factors causes full sterility, but they differ in chromosomal phenotypes. The meiotic chromosomes of *rad51* plants are highly fragmented, and those of *dmc1* plants appear intact but do not form bivalents due to a lack of COs. A recent study in Arabidopsis found physical interaction between RAD51, DMC1, SNI1, and ASAP1 [37]. However, in vitro competition experiments revealed that RAD51 attenuates the interaction of DMC1 with the two SMC5/6 complex subunits. Based on this and other experiments, the authors proposed that RAD51 inhibits SMC5/6 during the early meiotic prophase and thus allows for DMC1-mediated strand invasion. Such interaction could occur at ssDNA around the DSBs because all three factors, RAD51, DMC1, and SMC5/6 complex, have ssDNA binding affinity [46,47,48].

So far, the most attention has been paid to SMC5/6 complex activity in processing meiotic DSBs and aJMs. However, it seems important for successful chromosome segregation during meiosis. We found that Arabidopsis *nse2* plants produce approximately one-third of unreduced (diploid) microspores [40]. In some *nse2* meiocytes, all chromosomes remain in one pole. This phenotype is independent of SPO11-induced DSBs, as suggested by the analysis of *nse2 spo11* plants. Furthermore, the analysis of *nse2 osd1* meiotic products revealed 80% dyads and 20% monads. Because mutants in OMISSION OF SECOND DIVISION 1 (OSD1) skip the second meiotic division and produce dyads instead of tetrads [49], this *nse2 osd1* phenotype demonstrates that the non-reduction occurs mostly in meiosis I. However, meiosis II or both meiotic segregations can be omitted in a single *nse2* meiocyte, as suggested by the analysis of meiotic products from *nse2 qrt* plants (*qrt* mutation prevents separation of meiotic products), which included all classes, from tetrads to monads. The non-reduction in SMC5/6 complex mutants could be caused by several different defects. There could be an unnoticed population of SPO11-independent DSBs causing this problem. However, this option is less likely, as *nse2* and *sni1-1* plants do not show more RAD51 foci during the zygotene stage, and the phenotype is not linked with chromosome fragmentation [39,40]. Alternatively, we cannot exclude other DSB-independent types of DNA damage arising from, e.g., premeiotic replication or other chromatin-associated processes. In addition, some *nse2* meiocytes showed a disorganized and/or multipolar microtubule network. This might be caused by irregular chromosome positioning and/or structure, which was also observed in *nse2* meiocytes. Any of the above-described possibilities likely slow down cell division, which might not reach the meiotic checkpoint [50] on time, in which case, the cell with non-separated chromosomes moves to the next stage.

**Table 1 ijms-23-04503-t001:** The phenotypes of Arabidopsis SMC5/6 complex mutants. At—*Arabidopsis thaliana*, Zm—*Zea mays*, Os—*Oryza sativa*, *—under standard growth conditions. n.a.—not available, ins.—sequence insertion, del.—sequence deletion, ex—exon.

Gene Name	Gene ID	Mutant Allele	Stock ID/Mutation	Somatic Phenotype *	Meiotic Phenotype	Seed Phenotype	Reference
*AtSMC5*	At5g15920	*Atsmc5-1*	SALK_107583	Not viable	n.a.	Early embryo lethal (Type 1)	[31,45]
		*Atsmc5-2*	SALK_092081	Not viable	n.a.		
*AtSMC6A*	At5g07660	*Atsmc6a-2*	SALK_091553	WT-like	n.a.	WT-like	[51]
*AtSMC6B*	At5g61460	*Atsmc6b-4*	SALK_124719	WT-like	n.a.	WT-like	
		*smc6b-2*	SALK_135638	n.a.	Increased COs	n.a.	[39]
		*smc6a-2 smc6b-4*	SALK_091553 SALK_124719	Not viable	n.a.	Early embryo lethal (Type 1)	[51]
		*smc6a-1 smc6b-2*	SALK_009818 SALK_135638	n.a.	Increased COs	n.a.	[39]
*AtNSE1*	At5g21140	*Atnse1-1*	CS16151	Not viable	n.a.	Early embryo lethal (Type 1)	[52,53]
		*Atnse1-2*	SALK_136483	Not viable	n.a.	Early embryo lethal (Type 1)	
*AtNSE2/AtHPY2/AtMMS21*	At3g15150	*Atnse2-1/Athpy2-1/Atmms21-1*	Q115STOP	Short roots and stems, deformed leaves, stem fasciations, irregular branching, triploid individuals	Increased COs, fragmented and lagging chromosomes, anaphase bridges, monads to tetrads	Large seeds, cellularization defects, increased seed abortion (Type 2)	[21,22,40,54,55]
		*Atnse2-2/Athpy2-2/Atmms21-2*	SAIL_77_G06				
*AtNSE3*	At1g34770	*Atnse3-1*	GK-459F08	Not viable	n.a.	Early embryo lethal (Type 1)	[52,53]
		*Atnse3-2*	GK-534A03	Not viable	n.a.	Early embryo lethal (Type 1)	
*AtNSE4A*	At1g51130	*Atnse4a-1*	SALK_057130	Not viable	n.a.	Early embryo lethal (Type 1)	[56]
		*Atnse4a-2*	GK-768H08	Weakly delayed, triploid individuals	Increased COs, fragmented and lagging chromosomes, anaphase bridges and micronuclei	Large seeds, cellularization defects, increased seed abortion (Type 2)	
*AtNSE4B*	At3g20760	*Atnse4b-1*	SAIL_296_F02	WT-like	n.a.	WT-like	
		*Atnse4b-2*	GK-175D10	WT-like	n.a.	WT-like	
*AtASAP1*	At2g28130	*Atasap1*	GK-218F01	Strongly reduced growth, short roots	Fragmentated chromosomes	Almost sterile	[31,37]
*AtSNI1*	At4g18470	*Atsni1-1*	11 bp del., premature stop	Reduced growth, short roots	Fragmented chromosomes, increased Class II COs, dyads	Reduced fertility	[30,31,37,39]
		*Atsni1-2*	SAIL_298_H07	n.a.	Fragmented chromosomes	Almost sterile	[37]
		*Atsni1-3*	SAIL_34_D11	Short roots and stems, deformed leaves and triploid individuals	n.a.	Large seeds, cellularization defects, increased seed abortion (Type 2)	[40]
		*AtSNI1^Ler^*	I235V	WT-like	Recombination QTL, increased COs	n.a.	[39]
		*Atsni1-4* (as *Atsni1-2*)	14 bp del., premature stop	n.a.	n.a.	n.a.	
*ZmMMS21*	Zm00001d039007	*Zmmms21-1*	Mu ins., ex4	Slow growth, severely stunted plants, short roots, fewer leaves at maturity	n.a.	Small kernels, pitted surface, reduced embryo size and an underfilled endosperm, poor germination	[57]
		*Zmmms21-2*	Mu ins., ex6		n.a.		
		*Zmmms21-CR7*	33 bp del., 11 aa del. ex1		n.a.		
		*Zmmms21-CR1*	1 bp del., ex2, premature stop	Not viable to early somatic lethal	n.a.		
		*Zmmms21-CR3*	1 bp ins. ex2, new TSS producing a truncated protein	Not viable to early somatic lethal	n.a.		
		*Zmmms21-CR4*	1bp ins., ex1; 2 bp del., ex2, premature stop	Not viable to early somatic lethal	n.a.		
		*Zmmms21-CR6*	3 bp del., 1 aa del. ex1	Not viable to early somatic lethal	n.a.		
		*Zmmms21-CR2*	14 bp del., ex2, premature stop	Not viable	n.a.	Early embryo lethal	
		*Zmmms21-CR5*	6 bp del., 2 aa del. ex1; 8 bp del. ex2, premature stop	Not viable	n.a.	Early embryo lethal	
*OsMMS21*	LOC_ Os05g48880	*Osmms21*	05Z11BH79	Short roots, dwarf plants	n.a.	n.a.	[58]

Little is known about the role of the SMC5/6 complex in female meiosis. Arabidopsis NSE4A was found to be strongly expressed in the megaspore mother cell [56]. Furthermore, the presence of ovules without embryo sacs in *nse2* plants suggests occasional failure of female meiosis [40]. Another emerging role of the SMC5/6 complex might lie in securing chromosome stability in the meiosis of polyploid plants. This could be of high importance for plant breeding because many crops are extant polyploids. Polyploid meiosis differs in some respects from that of diploid meiosis [59,60]. A recent pioneer study on the reproductive development of autotetraploid (4×) Arabidopsis *nse2* plants revealed combined effects of polyploidy and SMC5/6 complex loss of function on meiotic phenotypes [54]. The 4× *nse2* plants showed greater tolerance to aneuploidy and its transmission through the male and (surprisingly) female meiosis. This is probably allowed by the presence of at least one full chromosome complement and a second chromosome complement with few either missing or excessive chromosomes. How the combination of tetraploidy and SMC5/6 complex loss of function affect the ratio of Class I and II COs or processing of aJMs is yet to be addressed.

## 3. Emerging Roles in Plant Gametophytic Development

The female and male meiosis produce mega- and microspores that undergo a series of cell divisions and differentiation in plants to form megagametophyte (embryo sac) and microgametophyte (pollen), respectively. These stages have not been extensively investigated concerning SMC5/6 complex functions (Table 1).

Several studies revealed shrunken pollen and reduced pollen viability in *nse2*, *nse4a-2*, *asap1*, and *sni1* [37,39,40,55,61]. In addition, there is less pollen germination and reduced pollen tube growth for nse2 [55]. However, these defects seem to be relics of problems arising during meiosis [40]. At the bicellular and tricellular stages of Arabidopsis pollen development, an abnormal number of nuclei and chromatin loss were occasionally observed. Whether these are also meiosis-derived defects and/or are caused by problems in the pollen mitotic divisions is unknown.

To the best of our knowledge, the morphology of the embryo sac has been analyzed only in *nse2* mutants of Arabidopsis and revealed a wide range of defects [40,55]. In the most extreme cases, the embryo sac was absent, which could result from an aborted female meiocyte. In addition, there were embryo sacs with variable numbers of nuclei, equally sized egg and central cell nuclei, or incorrect positions of the cells within the embryo sac. The latter problems could result from megagametophyte mitotic divisions by unequal segregation and possibly incorrect orientation of the mitotic spindles. The last problem might again (compared to lack of chromosome reduction in meiosis) suggest an effect of the SMC5/6 complex on microtubule network organization.

## 4. Direct and Indirect Effects on Seed Development

Seed formation starts with double fertilization, where one sperm nucleus fuses with the egg cell nucleus and the second sperm nucleus fuses with the central cell nucleus, giving rise to a diploid embryo and triploid endosperm, respectively [62]. Whereas the embryo represents a new plant generation, the functions of endosperm are manifold and include nourishing the embryo, sensing compatibility of the parental genomes, and, in some species, providing energy during germination. Endosperm development starts with several rounds of cell division without cell wall formation (syncytium). Several days after fertilization, endosperm cellularizes; this step is critical for successful seed development. Normal endosperm development and cellularization require a balanced dosage of maternal (m) and paternal (p) genomes, which is, by default, defined as a 2m:1p ratio [63]. Early studies in Arabidopsis revealed a class of mutants with unusually large seeds, disrupted endosperm cellularization, and arrested embryo development [64]. Genes underlying this phenotype were called TITAN, and some were mapped to subunits of cohesin and condensin complexes [9,65]. The presence of giant nuclei in their endosperm indicated problems in cell division. Recently, defects in seed development were described for the mutants in the SMC5/6 complex. Although there is a resemblance with the titan seeds, we propose that the SMC5/6 complex mutants display not one but at least two principal seed phenotypes (Type 1 and 2), differing as to their origin (Figure 2; Table 1).

Type 1 defective seeds correspond to the originally described EMBRYO DEFECTIVE (EMB) phenotype for SMC5 (EMB2782) and NSE1 (EMB1379). These abnormal seeds are initially of normal size, contain early-aborting embryos, and shrink during desiccation (Figure 2). This phenotype is inherited as a Mendelian recessive trait and was found in *nse4a-1*, all *smc5*, *nse1*, and *nse3* alleles, as well as in *smc6a-2 smc6b-4* double mutant (Table 1). For Arabidopsis SMC6 paralogs, the Type 1 seed phenotype is observable only in the double-mutant background because of the partial functional redundancy between SMC6A and SMC6B. Although the cause of the Type 1 seed phenotype is unclear, it strongly correlates with the signatures of genome instability, and we propose that these defects are directly caused by the deficient SMC5/6 complex function in seeds. Several DNA damage repair and cell cycle genes were upregulated in *nse1*, *nse3*, and *smc6a smc6b* seeds [51,52,53]. Cells activate checkpoints to arrest cell cycle progression, allowing sufficient time for DNA damage repair. The deficient DNA damage repair in these mutants may delay the cell cycle progression, cause growth inhibition, and finally lethality. The appearance of giant and heteromorphic endosperm nuclei in *nse1*, *nse3*, and *smc6a smc6b* mutant seeds indicates elevated endoreduplication instead of normal nuclear division [51,52,53]. The additional rounds of endoreduplication may delay the transition from proliferation to cellularization in the endosperm, thus arresting embryo development. NSE2 and SNI1 have been reported to inhibit E2F transcription factors in somatic tissues [66], suggesting that the SMC5/6 complex may contribute to cell cycle control. This also seems to be connected with auxin hormonal disbalance during embryo development as found in *nse1* and *nse3* seeds. Furthermore, many genes related to auxin signaling were downregulated in *nse1* and *nse3*, including members of *INDOLEACETIC ACID-INDUCED PROTEIN* (*IAA*), *AUXIN RESPONSE FACTOR* (*ARF*), and *YUCCA* (*YUC*) families [52]. This supports the role of the SMC5/6 complex in the regulation of normal auxin biogenesis, translocation, and transduction. However, the connection between the SMC5/6 complex and the auxin pathway might be rather indirect. Serious defects in DNA damage repair may lead to cell death. Both vacuolar programmed cell death (PCD) and necrotic PCD were observed in embryos of *nse1*, *nse3*, and *smc6a smc6b* seeds [51,53]. However, it is still not clear whether PCD occurs and is possibly initiated in the mutant endosperm.

The second aberrant seed phenotype (Type 2) of SMC5/6 complex mutants is hallmarked by large seed size, glossy surface, liquid endosperm, and an embryo reaching the heart or even later stages (Figure 2; Table 1). Such seeds are found in homozygous *nse2-1*, *nse2-2*, *nse4a-2*, and *sni1-3* plants [40,51,52,56]. Interestingly, Type 2 abnormal seeds are induced paternally [40], suggesting problems in the parental genome dosage. Whereas the double fertilization with reduced gametes produces a diploid embryo (1m:1p) and triploid endosperm (2m:1p), fertilization with unreduced sperm nuclei (as observed in nse2 plants) gives rise to a triploid embryo (1m:2p) and a tetraploid endosperm (2m:2p). Although direct evidence that the Type 2 seeds are the product of fertilization by unreduced sperm nuclei is missing, this hypothesis is indirectly supported by the improved seed viability in crosses of diploid *nse2* maternal plants with tetraploid wild-type paternal plants [40]. Hence, the Type 2 abnormal seed phenotype is most likely a consequence of aberrant meiosis when chromosomes fail to divide, and a diploid male gamete is produced, which leads to an incorrect parental dosage in the endosperm. Finally, the majority of such seeds die, but the minority can germinate and give rise to triploid offspring, as found for *nse2*, *nse4a-2*, and *sni1-3* plants [40].

This is the first observation of triploids among smc5/6 complex mutants, possibly because plants tolerate triploidy and aneuploidies better than many other eukaryotic groups. Arabidopsis triploid plants exhibit various degrees of sterility but can function as a bridge towards karyotypically more stable tetraploids [59]. However, newly formed autotetraploid *nse2* and *nse4a-2* plants were almost sterile and produced hexaploid and aneuploid progeny with the extra genome copies or chromosomes from both parents [54], indicating that the SMC5/6 complex is also essential for stable tetraploid plant development.

## 5. Importance of Crop Fertility

Current knowledge on the role of the SMC5/6 complex in plants is based on research on *Arabidopsis thaliana*, which is an excellent genetic model species representing plants with a small genome and low abundance of repetitive sequences [67]. However, most plants have larger genomes, which may be connected with a more complex genome organization and even greater necessity of activities performed by SMC complexes [68]. Therefore, it is exciting that the first studies on SMC5/6 complex functions in other plant species are emerging.

Maize (*Zea mays*) represents a moderately sized grass genome (2.4 Gbp/1C) and a very important crop [69]. Knocking out maize MMS21/NSE2 revealed two principal phenotypes [57]. First, the more severe mutations caused lethality in the seed stage or soon after germination. Second, the less severe mutants had slightly smaller seeds with a pitted surface, reduced embryo size, and often underfilled endosperm. This indicates that the function of the SMC5/6 complex to control seed development is conserved in plants. Such seeds had lower germination, and the seedlings grew more slowly, ultimately developing into severely stunted plants with shorter roots and fewer, shorter, and narrower leaves at maturity [57]. Another study focused on the rice somatic phenotypes of *OsMMS21*, the ortholog of *AtMMS21* [58]. The T-DNA insertion mutant had short roots and was dwarfish from the early tiller to the adult stages.

These pioneer studies suggest that the functions of the SMC5/6 complex is also relevant for agriculturally important traits in cereal and possibly other crops.

## Figures and Tables

**Figure 2 ijms-23-04503-f002:**
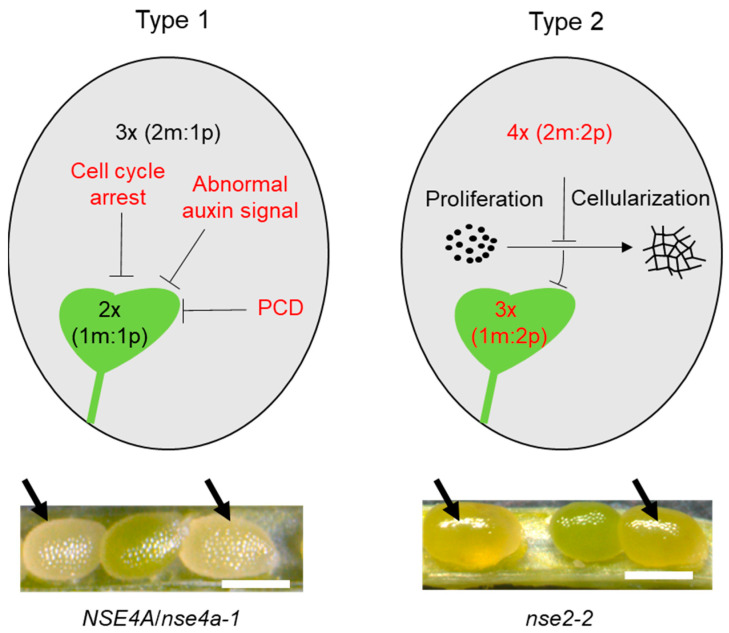
Two types of abnormal seed phenotypes were found in the SMC5/6 complex mutants. The ovals represent schematically drawn seeds with embryos in green and endosperm in gray. Mutant seed Type 1 has normal ploidy with diploid (2×, 1m:1p) embryo and triploid (3×, 2m:1p) endosperm, embryo aborts in early stages due to cell cycle arrest, abnormal auxin signal, and programmed cell death (PCD). The image below shows Arabidopsis seeds 13 days after pollination (DAP), with Type 1 seed indicated by the arrow. The mutant seed Type 2 has abnormal ploidy with triploid embryo (1m:2p) and tetraploid (4×) endosperm containing two maternal and two paternal genomes (2m:2p). An excess of paternal genome delays endosperm cellularization, inhibits embryo development, and frequently leads to seed abortion. Examples of Type 2 seeds are shown below and indicated by arrows. Type 2 seeds usually appear larger than average seeds, with a smooth surface. Bars = 500 μm.

## Data Availability

Not applicable.

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
