# Peer review of "Multiple Roles of SMC5/6 Complex during Plant Sexual Reproduction"

_ijms, 2022, doi:10.3390/ijms23094503_

Round 1
Reviewer 1 Report
The review is well structured and timely in the field.
Author Response
We would like to thank reviewer 1 for such great support.
Reviewer 2 Report
The manuscript “Multiple roles of SMC5/6 comlex during plant sexual reproduction” by Yang and Pecinka presents a literature review on the role of the SMC5/6 complex in plant meiosis and gamete development. The authors first present a brief overview of SMC5/6 complexes in general and in particular plants followed by an overview of meiosis before discussing the several studies that examined the role of SMC5/6 complex proteins in Arabidopsis. The authors follow this with relatively shorter sections on possible roles of the SMC5/6 complex in gametophyte and seed development in Arabidopsis and a short section on recent studies in maize and rice.
The manuscript is on an interesting topic and appears to represent a reasonably thorough review of the literature. The manuscript is generally well written but does contain a number of grammatical problems that can be distracting and sometimes make the manuscript confusing. One example is lines 182-184, which really don’t make sense. The manuscript is also longer than need be and could be shortened without losing any important information. It also contains a fair amount of speculation that should be removed. Likewise, it is not clear if gametophytic alterations observed in SMC5/6 complex subunit mutants are the direct result of the mutations or alterations resulting from meiotic defects. The authors favor the conclusion that SMC5/6 is required for gametophyte development, which it may be, but the evidence is not all that clear. For example: lines 67-71 and 168-171 could easily be removed. The figures are good and for the most part informative. In Figure 1, the authors should clarify what organism is being referred to.
Author Response
Response: We would like to thank reviewer #2 for the overall positive evaluation of our manuscript.
Reviewer 2: The manuscript is generally well written but does contain a number of grammatical problems that can be distracting and sometimes make the manuscript confusing. One example lines 182-184, which really don’t make sense.
Response: We carefully checked the manuscript, corrected typing errors, and modified some unclear parts throughout the text. We have rewritten the text on lines 182-184 to make it clearer. It now reads as follows: „Little is known on the role of the SMC5/6 complex in female meiosis. Arabidopsis NSE4A was found strongly expressed in the megaspore mother cell [52]. Furthermore, the presence of ovules without embryo sacs in nse2 plants suggested occasional failure of female meiosis [37].“
Reviewer 2: The manuscript is also longer than need be and could be shortened without losing any important information. It also contains a fair amount of speculation that should be removed. Likewise, it is not clear if gametophytic alterations observed in SMC5/6 complex subunit mutants are the direct result of the mutations or alterations resulting from meiotic defects. The authors favor the conclusion that SMC5/6 is required for gametophyte development, which it may be, but the evidence is not all that clear. For example: lines 67-71 and 168-171 could easily be removed.
Response: We agree that the published works should be as concise as possible and have somewhat shortened the manuscript based on the reviewer’s suggestions. We removed lines 67-71, 209-212, 221-223 and strongly reduced the text between lines 168-171.
The more „speculative“ parts reflect our current knowledge of the functions of the SMC5/6 complex in plants and formulate possible mechanisms that could be responsible for the observed phenotypes. We consider this as a valuable part of our review which somehow articulates testable hypotheses that can be confirmed or rejected over the coming years. However, the word „speculative“ was removed from the text.
Ad gametophytic defects. After reading the reviewer’s comments, we believe to have a very similar view on this situation. The role of SMC5/6 in female gametophyte development is well supported by two of our own studies but the role in male gametophyte development remains unclear. We removed lines 221-223 from the past dedicated to gametophytic defects which were possibly blurring this message.
Reviewer 2: The figures are good and for the most part informative. In Figure 1, the authors should clarify what organism is being referred to.
Response: We added information that the model is based on Arabidopsis.
Reviewer 3 Report
The SMC5/6 complex is evolutionarily conserved and plays diverse roles in DNA damage repair. Most previous studies on SMC5/6 was conducted in somatic cells. However, more and more recent papers reported its role in germline cells. This review summarized the progresses of plant SMC5/6 in sexual reproduction. The review was well-written and included the most updated studies. I strongly support it publication in IJMS and believe that it will be highly valuable to readers in this field.
- Line 11, 36, 61: “evolutionary”should be “evolutionarily”.
- Line 16: Please add a space between “is” and “attenuated”.
- Line 60: “there are two to three evolutionary non-conserved subunits......”. Rtt107 and Brc1 are not normally considered to be the subunits of the SMC5/6 complex.
- Line 132: " ase" shoud be “are”.
- Line 254: " thna" should be “than”.
- There are many formatting errors in the reference list (e.g., 3, 11, 18, 19, 20).
Author Response
We would like to thank reviewer 2 for this very positive evaluation of our manuscript.
In the revised version, we have performed corrections as suggested. We have done additional checks for typing errors and the English language.
Specific points:
Line 11, 36, 61: “evolutionary”should be “evolutionarily”.
Response: Corrected.
Line 16: Please add a space between “is” and “attenuated”.
Response: Corrected.
Line 60: “there are two to three evolutionary non-conserved subunits......”. Rtt107 and Brc1 are not normally considered to be the subunits of the SMC5/6 complex.
Response: We have softened this statement as follows: "In yeasts, it is Nse5 and Nse6 that interact with BRCT domain-containing factors Rtt107 or Brc1 [26–28]." Our rationale was that both Rtt107 and Brc1 contain BRCT domain. The same domain is part of the human SLF1 protein that is considered as the standard SMC5/6 complex subunit.
Line 132: " ase" shoud be “are”.
Response: Corrected.
Line 254: " thna" should be “than”.
Response: Corrected.
There are many formatting errors in the reference list (e.g., 3, 11, 18, 19, 20).
Response: Corrected.